# Hybrid pointer networks for traveling salesman problems optimization

**Ahmed Stohy[1], Heba-Tullah Abdelhakam[1], Sayed Ali[1], Mohammed Elhenawy[2], Abdallah A. Hassan[1], Mahmoud Masoud[2]\*, Sebastien Glaser[2], Andry Rakotonirainy[2]**

**1** Department of Computer and Systems Engineering, Minya University, Minya, Egypt, **2** Centre for Accident Research and Road Safety, Queensland University of Technology, Brisbane, Australia

\* Mahmoud.masoud@qut.edu.au

**Data Availability Statement:** All data, code and models files are available at https://github.com/AhmedStohy/Hybrid-Pointer-Networks.

## Abstract

In this work, we proposed a hybrid pointer network (HPN), an end-to-end deep reinforcement learning architecture is provided to tackle the travelling salesman problem (TSP). HPN builds upon graph pointer networks, an extension of pointer networks with an additional graph embedding layer. HPN combines the graph embedding layer with the transformer's encoder to produce multiple embeddings for the feature context. We conducted extensive experimental work to compare HPN and Graph pointer network (GPN). For the sack of fairness, we used the same setting as proposed in GPN paper. The experimental results show that our network significantly outperforms the original graph pointer network for small and large-scale problems. For example, it reduced the cost for travelling salesman problems with 50 cities/nodes (TSP50) from 5.959 to 5.706 without utilizing 2opt. Moreover, we solved benchmark instances of variable sizes using HPN and GPN. The cost of the solutions and the testing times are compared using Linear mixed effect models. We found that our model yields statistically significant better solutions in terms of the total trip cost. We make our data, models, and code publicly available https://github.com/AhmedStohy/Hybrid-Pointer-Networks.

## I. Introduction

Combinatorial optimization problems have garnered substantial attention from the theory and algorithm design community in recent decades as fundamental challenges in computer science and operations research. TSP in one fundamental combinatorial optimization problem that have been explored in the disciplines of logistics transportation, genomics, express delivery, and dispatching. TSP is often defined on a graph with a number of nodes, and it is essential to search through the permutation sequences of nodes for finding an optimal one with the shortest traveling distance.

Due to the many applications of the travelling salesman problem (TSP) in many areas, it has received significant attention from the machine learning community in the past years. However, the developed neural combinatorial optimization models are still in the infantry stage. Generalization is still an unresolved problem when it comes to dealing with many points

**Funding:** The authors received no specific funding for this work.

**Competing interests:** he authors have declared that no competing interests exist.

with high precision. The travelling salesman problem (TSP) is considered as one of the most significant and practical problems. Consider a salesman travelling to several areas; the salesman must visit each city just once while minimizing the total travel time. TSP is an NP-hard problem [1], which addresses the challenge of finding the optimal solution in polynomial time.

Exact algorithms, approximation algorithms, and heuristic algorithms are examples of traditional approaches for tackling NP-hard graph optimization problems [2]. Exact algorithms using the branch and bound framework can produce optimum solutions, but due to their NP-hardness, they are not suited for large-scale applications. Polynomial-time approximation algorithms can often produce quality-guaranteed solutions, although they have lower optimality guarantees than precise algorithms. The optimality guarantee may not exist at all for situations that are not amenable to a polynomial approximation approach. Furthermore, because of their high computing efficiency, heuristic algorithms are commonly employed, although they typically need adaptations and subject specialist understanding for a given situation. Heuristic algorithms frequently lack theoretical basis, all three groups of algorithms previously mentioned seldom take advantage of the common features among optimization problems, and thus frequently require the design of a new algorithm to solve a different instance of an even similar problem that is based on the same combinatorial structure, with the coefficient values in the objective function or constraints regarded as samples from the same basic distributions [3]. The use of machine learning methodologies has provided a silver lining in the form of a scalable solution for solving combinatorial problems with similar combinatorial structures.

Many approximation algorithms and heuristics, such as Christofides algorithm [4], local search [5], and the Lin-Kernighan heuristic (LKH) [6] have been developed to overcome the complexity of the exact algorithms which are guaranteed to yield an optimal solution but are frequently too computationally costly to be utilized in practice [7]. Many combinatorial optimization problems, such as a TSP has a graph structure [8], which may be easily described using the current graph embedding or network embedding techniques. The graph information is integrated in a continuous node representation in this method. Because of its great skills in information embedding and belief propagation of graph topology, the most recent development of graph neural network (GNN) may be applied in simulating a graph combinatorial problem [9]. This drives us to use a GNN model to handle combinatorial optimization problems, specifically TSP.

The pointer network [10], a seq2seq model [11], shows great potential for approximation solutions to combinatorial optimization problems such as identifying the convex hull and the TSP. It uses LSTM [12] as the encoder and an attention mechanism [13] as the decoder to extract features from city coordinates. It then predicts a policy outlining the next likely city by selecting a permutation of visited cities. The pointer network model is trained using the Actor-Critic technique [14].

Moreover, the attention model [15, 16], influenced by the Transformer architecture [13], tried to address routing problems such as the TSP and VRP. Graph pointer networks [17] extended the traditional pointer networks with an additional layer of graph embedding, this transformation achieved a better generalization for a large-scale problem, but the GPN model without 2-opt still struggling for finding the optimal solutions for small and large scale.

GPN has limited capabilities for tackling small-scale problems, and the suggested GNN employed in its architecture isn't the ideal encoder for determining point-to-point relationships. The work proposed in this paper begins with how the performance of graph pointer networks can be improved without changing much of the architecture; an extra encoder layer is added alongside the graph embedding layer to act as a hybrid encoder, and this gives the model the ability to achieve good results; this will be discussed in greater detail in the HPN section.

Extensive experimental work results show that the proposed technique significantly outperforms previous DL-based methods on TSP. The learnt model is more successful than standard hand-crafted rules in guiding the improvement process, and they may be further strengthened by simple ensemble methods. Furthermore, HPN generalizes rather well to a variety of problem sizes, starting solutions, and even real-world datasets. It should be noted that the goal is not to outperform highly optimized and specialized traditional solvers but to present a generalized model that can automatically learn good search heuristics on different problem sizes, which has a great value when applied to real-world problems.

In this work, we propose a deep reinforcement learning model trained via Actor-critic. Our architecture is based on a pointer attention mechanism that outputs nodes sequentially for action selection. We introduce a reinforcement learning formulation to learn a stochastic policy of the next promising solutions using a hybrid architecture, incorporating the search's history information by keeping track of the current best-visited solution. Our results show that we can learn policies for the Euclidean TSP that achieve the state-of-the-art solution compared with previous work in RL-based Models. Moreover, our approach can achieve state of the art results compared with previous deep learning methods based on construction [10, 15, 18–20] and improvement [21] heuristics.

## II. Travelling salesman problem (TSP)

TSP is a classic example of a combinatorial optimization problem that has been used in data clustering, genome sequencing, as well as other fields. TSP problem is NP-hard, and several exact, heuristic and approximation algorithms have been developed to solve it. In this paper, TSP problems are assumed to be symmetric. The symmetric TSP is regarded as an undirected graph.

### 1. Asymmetric vs. symmetric TSP

The distance between two cities in the symmetric TSP is the same in each opposite direction, producing an undirected network. This symmetry cuts the number of alternative solutions in half. Paths may not exist in both directions in the asymmetric TSP, or the distances may be different, resulting in a directed graph.

### 2. Directed vs. undirected graphs

Edges in undirected graphs do not have a direction. Each edge may be travelled in both directions, indicating a two-way connection. The edges of directed graphs have a direction. The edges represent a one-way connection, as each edge may only be travelled in one direction.

A full undirected graph can be defined as $C = (V,E)$ where V is the vector of vertices of graph C, and E is the vector of edges between these vertices. In this study, the TSP's graph is complete, so every node has an edge to each of the other vertices in the graph.

$$C \text{ is symmetric if } (\forall i, j : e_{ij} = e_{ji}),$$

In the context of this paper, $e_{ij}$ equals the distance between the vertices $i$ and $j$. Given a set $V$ of cities $n$ in a two-dimensional space, the objective is to find the optimal Hamiltonian path that minimizes the total tour length [20]:

$$L(\pi|V) = \left\| v_{\pi(n)} - v_{\pi(1)} \right\|_2 + \sum_{i=1}^{n-1} \left\| v_{\pi(i)} - v_{\pi(i+1)} \right\|_2 \tag{0.1}$$

Where $\|.\|_2$ is $\ell_2$ norm and $\pi$ denote as a tour.

## III. Methods

In this section, we will describe the learning algorithm used to train the proposed model. Furthermore, we describe the statistical model used to test whether our proposed model outperforms the GPN.

### 1. Reinforcement learning (RL)

Reinforcement learning (RL) is the process of learning what to perform to increase the expected numerical reward signal. The agent isn't instructed which actions to perform but must experiment to determine which acts offer the greatest expected reward. To begin, we will define the notation used to represent the TSP as a reinforcement learning problem. Let $S$ *be* the state space and $A$ the action space. Each state $s_t \in S$ is defined as the set of all previously visited cities. Action $a_t \in A$ is defined as the following selected city from the group of possible cities; our model is considered a sequential one that, given an instance $a_t$ (selected input city) outputs a probability distribution over the next candidates from the remaining cities that have not been chosen. We can define our policy as:

$$P_\theta(\pi|x) = P[A_t = a | S_t = s], \tag{0.2}$$

From which we can sample to obtain a tour $\pi$. In order to train our model, we define the loss [15]:

$$\mathcal{L}(\theta|s) = \mathbb{E}_{P_\theta(\pi|x)}[L(\pi)], \tag{0.3}$$

Where $L(\pi)$ is the cost of the tour that we are attempting to minimize. Recall the REINFORCE's [22] equation with baseline which is an extension from policy gradient algorithm [23]:

$$\nabla J \approx \mathbb{E}[(L(\pi) - b(s))\nabla \log \pi(a|s)]. \tag{0.4}$$

Where $b(s)$ is the baseline subtracted from the cost to eliminate the policy gradient variance. The optimal baseline is one that lowers variation as much as possible while simultaneously speeding up the training process. As a result, we employ the approach given by [15]:

```
Algorithm 1. REINFORCE with Rollout Baseline [15]
1: input: number of epochs E, steps per epoch T, batch size B, signifi-
cance α
2: init θ, θ^BL ← θ # initialize network parameters
3: for epoch = 1,..., E do
4: for step = 1,..., T do
5: s_i ← RandomInstance () ∀ i ∈ {1,..., B} # Generate Random Instances
6: π_i← SampleRollout (s_i,p_θ) ∀ i ∈ {1,..., B} # Sample from policy
7: π_i^BL ← GreedyRollout (s_i, p_θ^BL) ∀ i ∈ {1,..., B} # Greedy selection from
policy
8: ∇L ← Σ_{i=1}^B L(π_i) − L(π_i^BL)∇_θ log p_θ(π_i) # Loss calulation ((Sampling cost–BL
cost)* logprob)
9: θ ← Adam (θ, ∇L) # Optimizer step
10: end for
11: if OneSidedPairedTTest (p_θ, p_θ^BL) < α then # Check if Actor is better
than critic with margin α
12: θ^BL ← θ # Transfer Actor's weights into Critic
13: end if
14: end for
```

## 2. Linear mixed effect models

Linear mixed effect models (LMEs) are tools used to test whether a significant relationship exists between the dependent variable (response) and independent variables (regressors). LMEs are developed to enable the analysis of dependent data by introducing random variables (i.e., random effects) at the lower levels of the model. For example [24], different algorithms returned solutions to the same instance. To capture the correlation between the solutions of the same instance, a random effect is introduced at the instance level. The formula of the mixed model is shown below

$$y = X\beta + Zu + \varepsilon$$

Where

$y$ is the response vector

$X$ is the design matrix for fixed factors

$\beta$ is the coefficients of fixed effect regression

$Z$ is the random effects design matrix for random factors

$U$ is the vector of random effects

$\varepsilon$ is the residuals

In the linear mixed model, the fixed factor $x_i$'s null hypothesis is that $x_i$ does not significantly explain some of the variability of the response $y$ (i.e., $\beta_i = 0$). For the purposes of this paper, the significance level, $\alpha$, is set at 0.05. To determine if a particular fixed factor significantly affects the response, the p-value corresponding to this fixed factor in the estimation table is compared with the 0.05 significance level.

## IV. Hybrid pointer network (HPN)

HPN is inspired by the Graph pointer network (GPN). GPN is a modified variant of the classic pointer network (PN).

Graph pointer networks have been used to tackle TSP. Building on this approach, in this paper:

- The graph embedding layer is combined with the transformer's encoder to produce multiple embeddings for the feature context.

- An extra decoder layer is added to operate as a multi-decoder structure network to improve the agent's decision-making process throughout the learning phase.

- Finally, we switch our learning algorithm from a central self-critic [25] to an actor-critic one as suggested by Kool [15].

The GPN adds graph embedding layer above the pointer network, allowing the model to figure out the complicated relationships between graph nodes in large-scale problems. However, it still struggles to find a globally optimal strategy for TSP problems. This study proposes extending the network architecture to converge to a better policy for small, medium, and large sizes. The proposed HPN is shown in Fig 1. HPN consists of a mixture of several encoder's architecture and multi decoder based on the attention concept.

### 1. Hybrid encoder

As illustrated in Fig 2, the proposed encoder consists of two parts: the hybrid context encoder, which encodes the Feature vector into two contextual vectors and the point encoder, which encodes the currently selected city by LSTM. Two different encoders are employed for the hybrid context encoder. The first encoder is a typical transformer encoder with multi-head

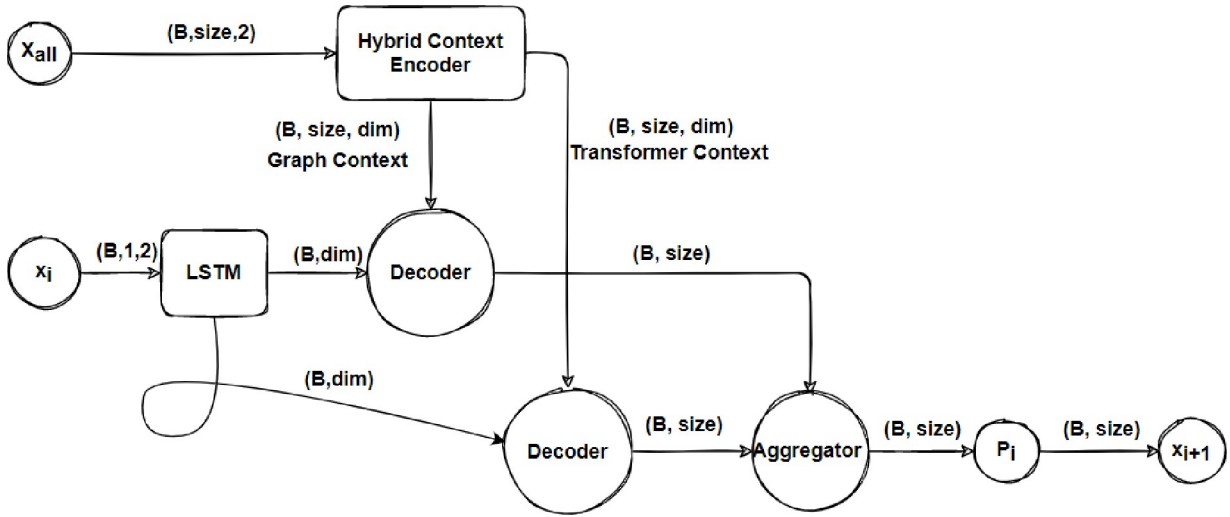

**Fig 1. Architecture of HPN which combining a hybrid context encoder with a multi-attention decoder.**

attention and residual connection with batch normalizing layer, the transformer's encoder equations with a single head are [26]:

$$H^{enc} = H^{l=L^{enc}} \in R^{(n+1) \times d} \tag{0.5}$$

$$H^l = softmax\left(\frac{Q^l K^{l^T}}{\sqrt{d}}\right) V^l \in R^{(n+1) \times d}, \tag{0.6}$$

$$Q^l = H^l W_Q^L \in R^{(n+1) \times d}, W_Q^l \in R^{d \times d}, \tag{0.7}$$

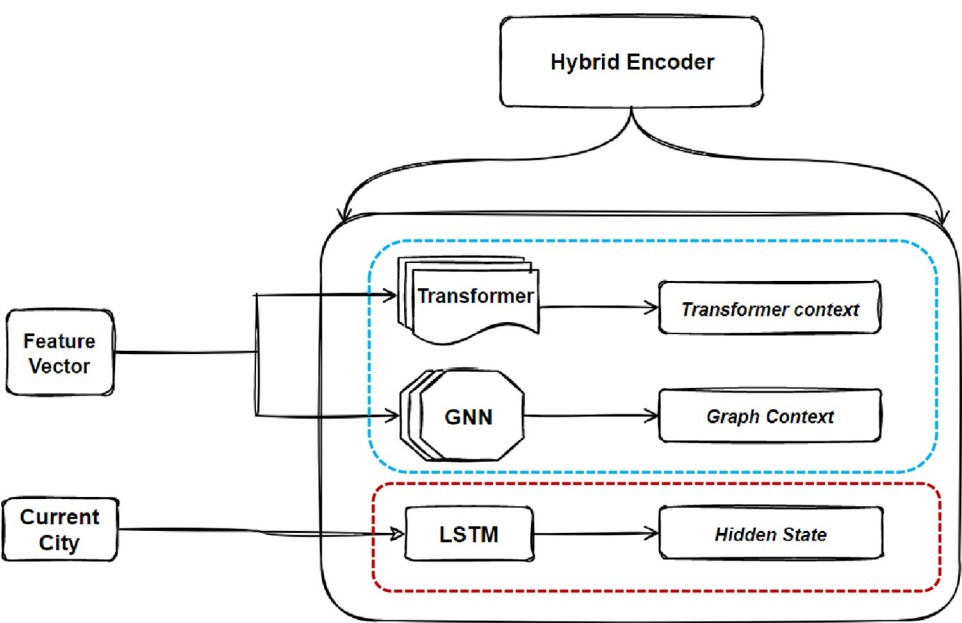

**Fig 2. Hybrid encoder consists of transformer's encoder and graph embedding layer as a hybrid context encoder, (blue dotted box) for the hybrid context encoder and (red dotted box) for the point encoder.**

$$K^l = H^l \, W_K^L \in R^{(n+1) \times d}, \, W_K^l \in R^{d \times d}, \tag{0.8}$$

$$V^l = H^l \, W_V^L \in R^{(n+1) \times d}, \, W_V^l \in R^{d \times d} \tag{0.9}$$

Where $W_Q^L$, $W_K^L$ and $W_V^L$ are learnable parameters, $H^{enc}$ is a matrix contains the encoded nodes, $Q^l$, $K^l$ and $V^l$ are a query, key and value of the self-attention.

The second one is the graph embedding layer. The graph embedding layer context is acquired by directly encoding the context vector obtained from coordinates of cities. Because we are only considering symmetric TSP, the graph is full. As a result, the graph embedding layer can be written as [17]:

$$X^l = \gamma X^{l-1} W_g + (1 - \gamma) \varphi_\theta \left( \frac{X^{l-1}}{|N(i)|} \right) \tag{0.10}$$

Where $X^l \in R^{N \times d_l}$, and $\varphi_\theta : R^{N \times d_{l-1}} \to R^{N \times d_l}$ is the aggregation function, $\gamma$ is a trainable parameter, $W_g \in R^{d_{l-1} \times d_l}$ is trainable weight matrix and $N(i)$ the adjacency set of node i.

For the point encoder which encodes the currently selected city, each city coordinates $x_i$ (i.e. $(x_{i1}, x_{i2})$) is embedded into a higher dimensional vector $\hat{x} \in R^d$, where d is the hidden dimension. An LSTM then encodes the vector $\hat{x}$ for the current city $x_i$. The hidden variable $x_i^h$ of the LSTM is passed to both the decoder of the current stamp and the encoder of the next time stamp.

## 2. Multi-decoder

To begin the decoding phase, a placeholder is added for the first iteration of the decoding to select the best location to start the tour, the decoder is based on the attention mechanism of a pointer network and outputs the pointer vector $u_I$, which is then sent through a Softmax layer to build a distribution across the following candidate cities. The attention mechanism and the pointer vector $u_I$ are defined as follows [17]:

$$u_i^{(j)} = \begin{cases} V^T . \tanh(W_r r_j + W_q q) & \text{if } j \neq \sigma(k), \forall k < j, \\ -\infty & otherwise, \end{cases}$$

Where $u_i^{(j)}$ is the j-th entry of the vector $u_i$, $W_r$ and $W_q$ are trainable parameters, q is the query vector from the hidden state of the LSTM, is a reference vector containing the contextual information from all cities.

The encoded context from the transformer's encoder is used as a reference for the first decoder layer and the context obtained from the graph embedding layer is used as the reference for the second decoder layer, as illustrated in Fig 1.

Then we'll have an attention vector for each decoder layer, and we'll need to figure out how to aggregate them. For the aggregator Function, four distinct procedures may be used to determine the distribution policy throughout the candidate cities:

- The first option is to add the two attention vectors from each decoder layer, which are provided by:

$$\pi_\theta(a_i | s_i) = p_i = softmax(u_i^1 + u_i^2)$$

- The second option is to take the maximum value between these two vectors, which is indicated as follows:

$$\pi_\theta(a_i|s_i) = p_i = softmax(\max(u_i^1, u_i^2))$$

- The third option is to take the mean as follows:

$$\pi_\theta(a_i|s_i) = p_i = softmax(average(u_i^1, u_i^2))$$

- The final option is to concatenate both of them and feed the concatenated vector into a single embedding layer, letting the model to decide how to aggregate them; we can describe this notion as follows:

$$\pi_\theta(a_i|s_i) = p_i = softmax(\vartheta_\theta(cat(u_i^1, u_i^2))).$$

Where $\vartheta_\theta : R^{N \times 2} \rightarrow R^{N \times 1}$ is the aggregation function. In the result section, we displayed the outcome for each one of them.

Fig 1 illustrates the model operations. We feed the network a tensor of input nodes in this problem the input nodes contain four features as previously illustrated so the input dimeson will be (batch-size, problem-size, number-of-feature), we feed these nodes into the hybrid context and will get two contextual vectors one from the transformer's encoder and the other from the graph encoder, then for the first decoding stamp we feed the placeholder to the pointer encoder for learning the best possible location. Finally, we feed the contextual vectors with the hidden states from the pointer encoder to our decoder, which is a simple attention layer, and aggerate the two-attention vectors using the sum operation. For clarity, we employ two decoder layers, one for the context vector of the graph and the other for the context vector of the transformer.

## V. Experiments

In our experiments, the city/node coordinates are independently and randomly drawn from a uniform distribution $x \sim U(0, 1)$. In each epoch, the training data is generated on the fly. The hyperparameters provided in Table 1 are used in the following experiments.

### 1. Small-scale experiments

We begin our experiments with a difficult barrier: which aggregator function between the previously described ones will assist our model in achieving better results? Indeed, it is difficult to answer this question without experimenting all of them; we examined the above-mentioned suggestions for this component and recorded the training performance results. Fig 3 illustrate these results.

**Table 1. Hyperparameters used for training.**

| Parameter | Value | Parameter | Value |
|---|---|---|---|
| Graph Embedding Layer | 3 | Learning rate | 1e-4 |
| Transformer Encoder Layer | 6 | Batch size | 512 |
| Feed-forward dim | 512 | Training steps | 2500 |
| Optimizer | Adam | Tanh clipping | 10 |

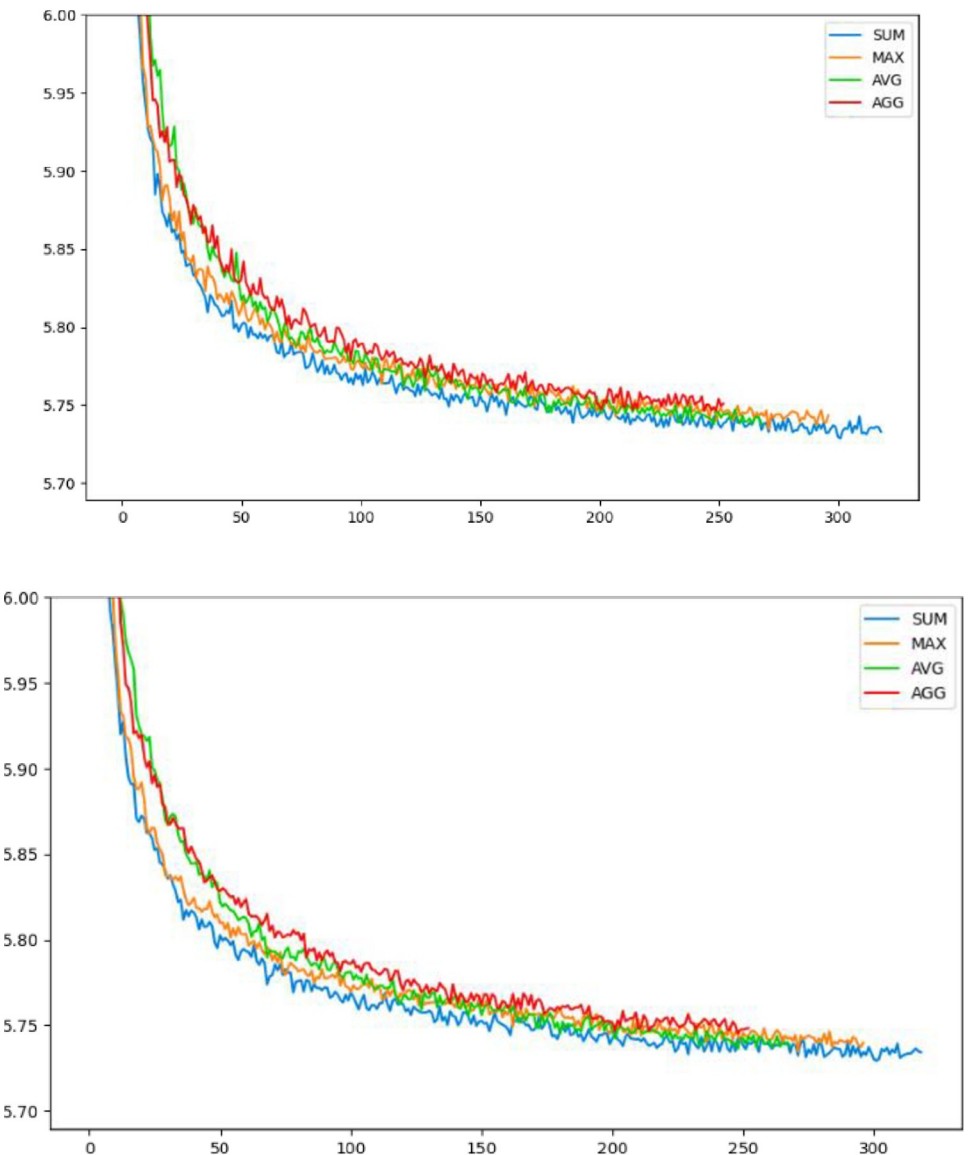

**Fig 3.** Training performance for the actor (on Top) and the critic (on Bottom) where the total tour length on the y-axis and the number of epochs on x-axis indicating that when we apply the sum operation between the two attention's vectors, the model converges a little fast compared with the others.

We can conclude from the above figure that the summation has excellent performance at first, but by the middle of training, the average has caught it, the maximum and the single-layer aggregation have a little higher result, so we decide to stop examining them.

We use 50 nodes travelling salesman problem (TSP50) instances to train our HPN model to tackle the small-scale. TSP50's average training time for each epoch is 19 minutes while utilizing one NVIDIA Tesla P100 GPU instance. We compare the performance of our model on small-scale TSP to earlier studies such as Graph pointer networks, the Attention Model, the pointer network, s2v-DQN [3], the Transformer Network [26] and other heuristics, e.g. 2-opt heuristics, Christofides algorithm and random insertion. The results are shown in Fig 4 which compares the approximate tour length to the optimal solution on 10k instances. A small number indicates a better result.

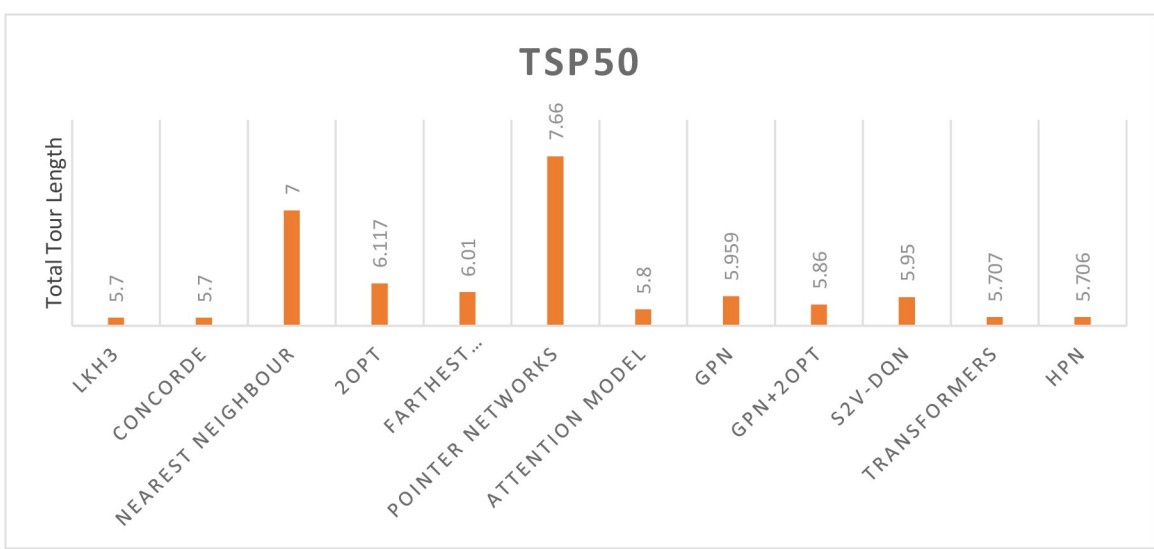

**Fig 4. Comparison of TSP50 results.**

As demonstrated in Fig 4 the optimal solution obtained from CONCORDE, LKH-3 heuristic is an extension of LKH-2 for solving constrained traveling salesman and vehicle routing problems, our HPN model surpasses the current existing models and achieves the state-of-the-art solution for TSP50. Our model outperforms the graph pointer network by a wide margin, enhancing its performance for TSP50 from 5.959 to 5.706 without utilizing 2-opt, which is a success for the hybridization concept.

## 2. Large-scale experiments

For achieving the best possible generalization out of our model, instead of just using the cities' coordinates as a context for both the graph encoder and the transform encoder, we replace it

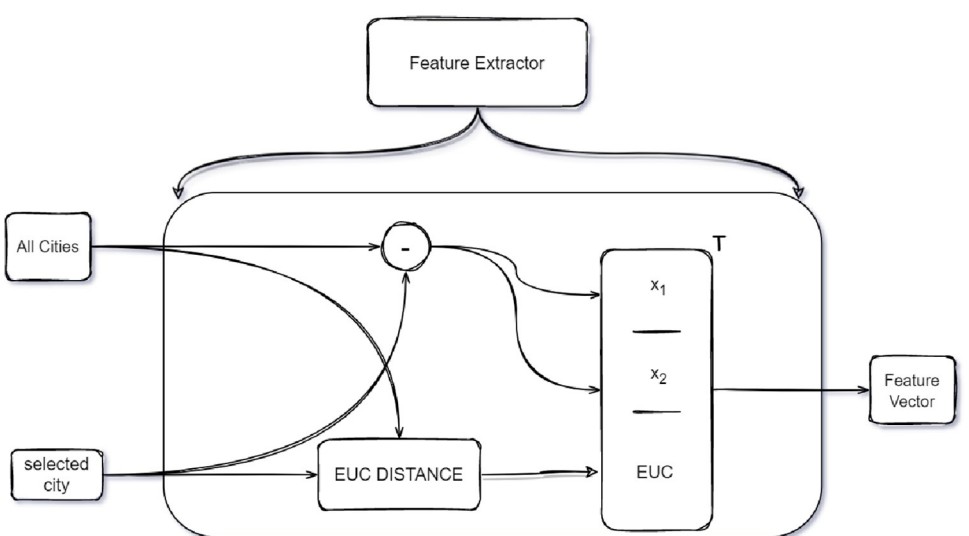

**Fig 5. The feature extractor architecture combines both vector context with the Euclidian distance and outputs a Feature vector.**

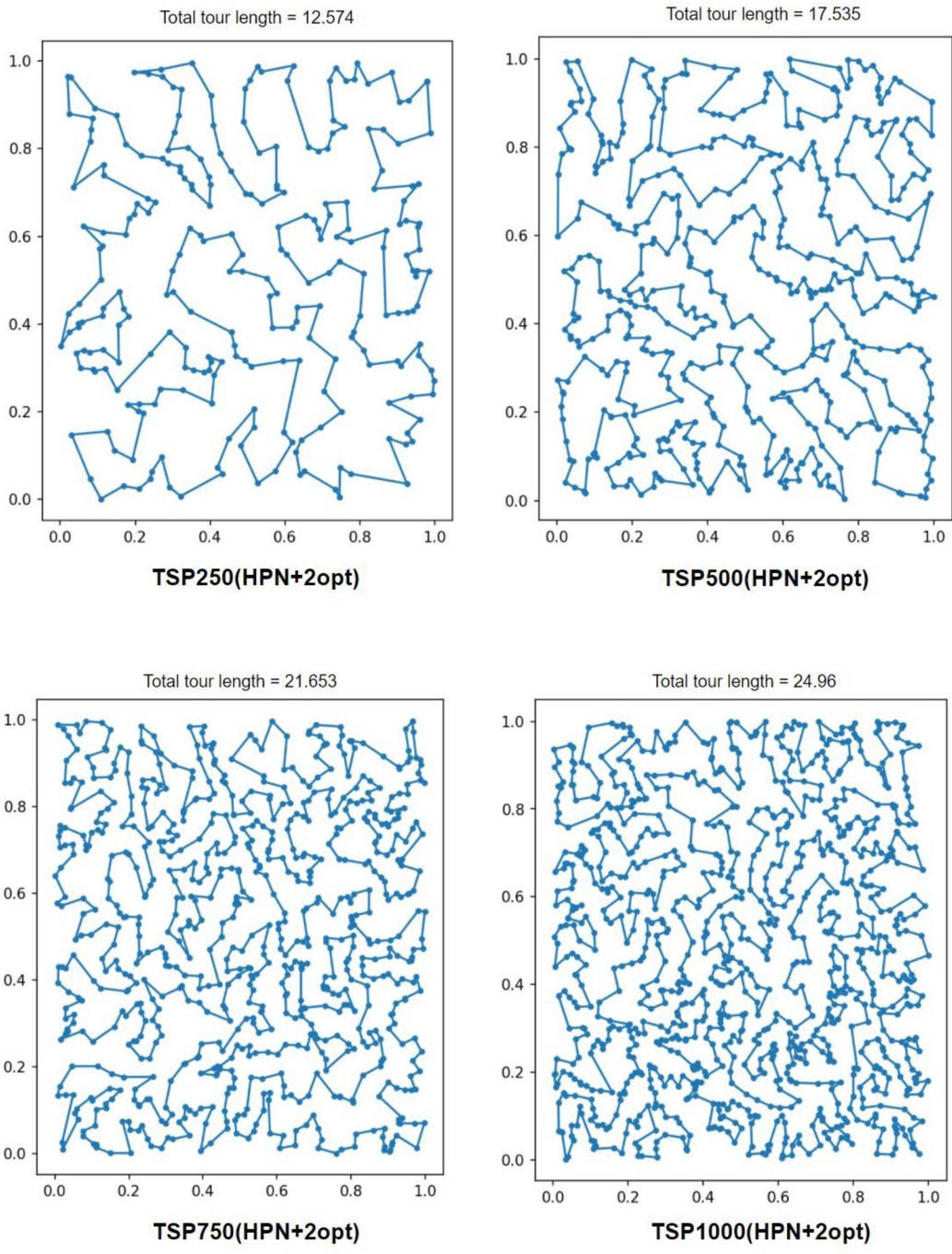

**Fig 6. Sample tours for TSP250-500-750-1000 solved by HPN+2-opt.**

with a feature context that accelerates the training convergence for the large-scale problems. The feature context includes the vector context previously used by [17] concatenated with the Euclidean distance, where the vector context is just a subtraction operation between the coordinates of the currently selected city with the others.

Our feature extractor component does this job as illustrated in Fig 5. It is essential in the proposed HPN model since, it extracts the most relative information and feeds it to the

**Table 2. TSP's result using hybrid pointer network model (HPN) vs baselines.** Each result is obtained by averaging on 1000 random TSP instances for larger instance and 10k instances for TSP50. Obj is the total tour length and the time reported is for solving 1k instances for larger TSP instances and 10k for TSP50.

| Method | TSP50 | | TSP250 | | TSP500 | | TSP750 | | TSP1000 | |
|---|---|---|---|---|---|---|---|---|---|---|
| | Obj. | Time | Obj. | Time | Obj. | Time | Obj. | Time | Obj. | Time |
| LKH3 | 5.70 | 300s | 11.893 | 9792s | 16.542 | 23070s | 20.129 | 36840s | 23.130 | 50680s |
| Concorde | 5.70 | 120s | 11.89 | 1894s | 16:55 | 13902s | 20.10 | 32993s | 23.11 | 47804s |
| Nearest Neighbor | 7.00 | 0s | 14.928 | 25s | 20.791 | 60s | 25.219 | 115s | 28.973 | 136s |
| 2-opt | 6.117 | 7.92s | 13.253 | 303s | 18.600 | 1363s | 22.668 | 3296s | 26.111 | 6153s |
| Farthest Insertion | 6.01 | 2s | 13.026 | 33s | 18.288 | 160s | 22.342 | 454s | 25.741 | 945s |
| OR-Tools (Savings) | – | – | 12.652 | 5000s | 17.653 | 5000s | 22.933 | 5000s | 28.332 | 5000s |
| OR-Tools (Christofides) | – | – | 12.289 | 5000s | 17.449 | 5000s | 22.395 | 5000s | 26.477 | 5000s |
| Pointer Net | 7.66 | – | 14.249 | 29s | 21.409 | 280s | 27.382 | 782s | 32.714 | 3133s |
| Attention Model | 5.80 | 2s | 14.032 | 2s | 24.789 | 14s | 28.281 | 42s | 34.055 | 136s |
| GPN | 5.959 | 1.75s | 13.679 | 32s | 19.605 | 111s | 24.337 | 232s | 28.471 | 393s |
| GPN+2opt | 5.867 | 6.5s | 12.942 | 214s | 18.358 | 974s | 22.541 | 2278s | 26.129 | 4410s |
| s2v-DQN | 5.95 | – | 13.079 | 476s | 18.428 | 1508s | 22.550 | 3182s | 26.046 | 5600s |
| Transformers (Gr.) | 5.707 | 13.7s | 14.60 | 4s | 23.63 | 10s | 30.77 | 15s | – | – |
| HPN (Gr.) **ours** | 5.706 | 0.36s | 13.44 | 16s | 18.94 | 48s | 23.15 | 100s | 26.64 | 168s |
| HPN+2opt **ours** | – | – | 12.78 | 315s | 17.95 | 1460s | 21.95 | 3405s | 25.21 | 6480s |

encoder as a context. "Suppose that $X_i = [x_i^T, \ldots, x_i^T]^T \in R^{Nx2}$ is a matrix with identical $N$ rows. We define $\bar{X}_t = X - X_i$ as the vector context. The j-th row of $X_i$ is a vector pointing from node $i$ to node $j$ and X is the matrix that contains coordinates of all cities. We expanded this

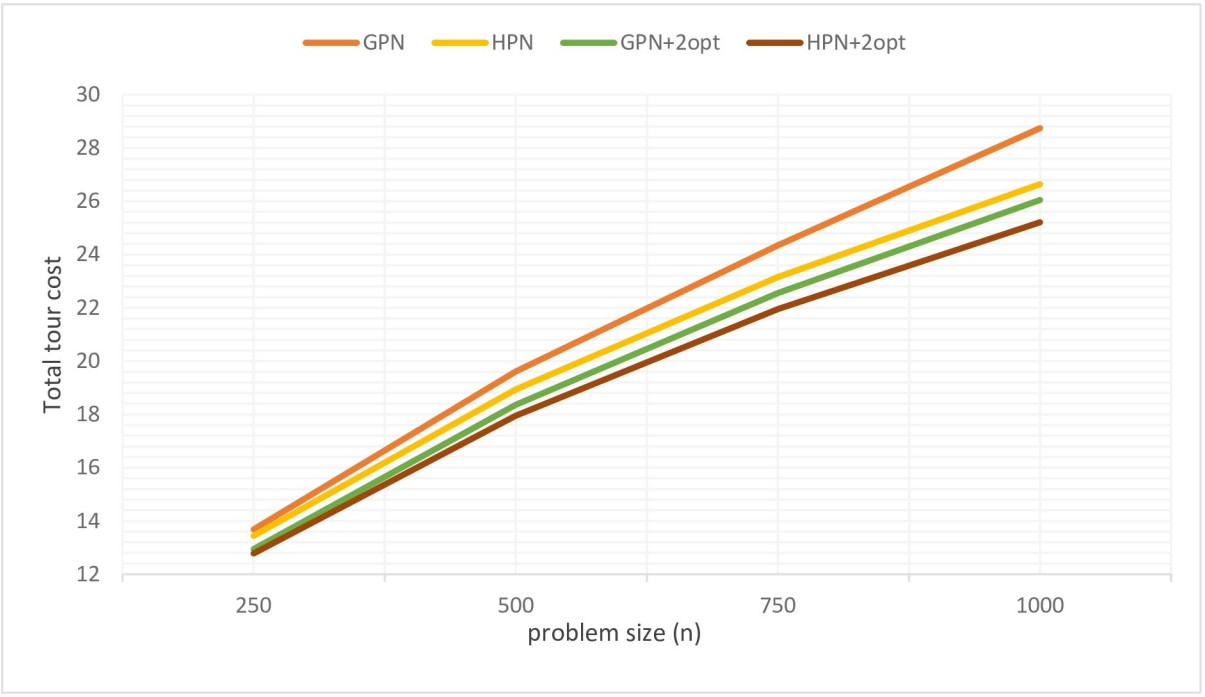

**Fig 7. Large-scale results from GPN, HPN, GPN+2opt, and HPN+2opt demonstrate that the gap between our model and GPN increases as the problem size increases.**

**Table 3. Evaluation on real world TSPLIB dataset using HPN and HPN+2opt.**

| Benchmark | HPN | | HPN+2opt | | Un-normalized tour length HPN+2opt in km | GPN | | GPN+2opt | | Un-normalized tour length GPN +2opt in km |
|---|---|---|---|---|---|---|---|---|---|---|
| | Obj. | Time | Obj. | Time | | Obj. | Time | Obj. | Time | |
| rd400 | 17.0 | 2s | **16.6** | 2.3s | **16538** | 18 | 1s | 16.9 | 2.1s | 16827 |
| gr431 | 10.4 | 1.3s | **9.9** | 1.8s | **2473** | 11.3 | 1s | 10.3 | 2.3s | 2689 |
| d493 | 13.4 | 1.3s | **11.2** | 3.6s | **38404** | 12.9 | 1.1s | 11.7 | 2.3s | 40098 |
| att532 | 14.8 | 1.4s | **13.2** | 3.4s | **96916** | 14.9 | 1.1s | 13.5 | 3.4s | 1029393 |
| pa561 | 18.0 | 2.8s | **17.3** | 5.3s | **16357** | 19 | 1.3s | 17.9 | 2.8s | 16971 |
| u574 | 18.3 | 1.4s | **17.2** | 3.6s | **41323** | 22.7 | 1.6s | 17.5 | 3.4s | 42874 |
| d657 | 16.3 | 1.8s | **15.2** | 5s | **53921** | 17.8 | 1.6s | 15.8 | 4.1s | 55692 |
| gr666 | 14.5 | 1.7s | **13.5** | 5.3s | **3731** | 15.9 | 1.67s | 13.9 | 5.5s | 3840 |
| u724 | 22.5 | 2.8s | **21.1** | 5.4s | **48742** | 22.7 | 1.5s | 21.4 | 4.7s | 49105 |
| rat783 | 25.1 | 2.2s | **24.7** | 4.4s | **10887** | 26.9 | 1.9s | 25 | 5.1s | 11015 |
| dsj1000 | 18.8 | 3s | **17.2** | 12s | **20715222** | 20.1 | 2.1s | 17.5 | 11.6s | 21020136 |
| u1060 | 23 | 2.8s | **20** | 11.4s | **286109** | 23.6 | 2.3s | 21 | 11.1s | 289678 |
| d1291 | 19.6 | 4s | **16.6** | 17s | **57713** | 20.1 | 2.6s | 16.7 | 20.1s | 57964 |
| nrw1379 | 27.2 | 4s | **25.8** | 15s | **61163** | 30.4 | 3.1s | 27.1 | 19.8s | 64005 |
| u1432 | 33.9 | 3.8s | **32.8** | 14.53s | **166739** | 36.2 | 2.9s | 33.6 | 16.3s | 170875 |
| vm1748 | 28.8 | 5s | **24.9** | 38s | **386483** | 29.3 | 3.8s | 25.9 | 31.15s | 395392 |
| rl1889 | 27 | 6s | **23.2** | 39s | **365362** | 29.6 | 4s | 23.6 | 34.1s | 373652 |
| u2152 | 36.9 | 6s | **32.3** | 43s | **75754** | 38.7 | 4.4s | 32.7 | 42.5s | 77622 |
| pr2392 | 38 | 7s | **35.2** | 70s | **426199** | 42.8 | 5.3s | 36.2 | 57.4s | 436979 |
| pcb3038 | 46.1 | 10s | **43.** | 117s | **154445** | 52.7 | 6.5s | 44 | 105s | 156771 |
| nu3496 | 26.4 | 9s | **23.8** | 144s | **105314** | 32.6 | 8.1s | 24.3 | 154s | 107965 |
| fl3795 | 16.4 | 2s | **14.7** | 170s | **30786** | 23.7 | 8.4s | 14.5 | 258.7s | 30402 |
| fnl4461 | 29.9 | 15s | **46.7** | 220s | **203653** | 59 | 9.3s | 49 | 166.7s | 214703 |
| ca4663 | 26.2 | 14s | **23.8** | 248s | **1585498** | 37.2 | 10.1s | 24.3 | 273.7s | 1621202 |
| rl5934 | 43.9 | 22.6s | **37.9** | 501s | **630300** | 54.8 | 13s | 39.6 | 426.3s | 645335 |
| tz6117 | 46 | 16s | **40.3** | 431s | **434403** | 55.6 | 12.7s | 41.6 | 409.1s | 448478 |
| eg7146 | 21.7 | 19.3s | **19** | 440s | **187376** | 35.3 | 16.7s | 19.2 | 633.3s | 189608 |
| pla7397 | 51.8 | 26s | **42.6** | 906s | **25169496** | 71.1 | 16.3s | 44.5 | 598.4s | 26287346 |
| ym7663 | 32.8 | 21s | **30.3** | 594s | **281001** | 57.7 | 16.6s | 30.8 | 792.5s | 286558 |
| ei8246 | 58.3 | 22s | **45** | 585s | **226498** | 75.5 | 18.1s | 55.8 | 737.4s | 233699 |
| ar9152 | 40.9 | 26s | **36.9** | 834s | **972758** | 56.5 | 19.7s | 38.2 | 1009.3s | 1002739 |
| ja9847 | 28.6 | 28s | **24.4** | 1090s | **542887** | 41.5 | 21.4s | 24.7 | 1438.5s | 550364 |
| fi10639 | 56 | 49s | **51.5** | 1206s | **573948** | 81.6 | 24.2s | 52.8 | 1238.6s | 5888001 |

**Table 4. The fixed effects coefficients of the model explaining the tour cost in terms of the HPN/GPN and the problem size.**

| Name | Estimate | SE | tStat | DF | pValue | Lower | Upper |
|---|---|---|---|---|---|---|---|
| (Intercept) | 16.01394 | 2.47475 | 6.470932 | 65 | < .0001 | 11.07152 | 20.95636 |
| size | 0.003976 | 0.000526 | 7.557891 | 65 | < .0001 | 0.002926 | 0.005027 |
| MODEL_1 | 6.485472 | 1.229939 | 5.273005 | 65 | < .0001 | 4.029116 | 8.941829 |

**Table 5. Fixed effects coefficients of the model explaining the testing time in terms of the HPN/GPN and the problem size.**

| Name | Estimate | SE | tStat | DF | pValue | Lower | Upper |
|---|---|---|---|---|---|---|---|
| (Intercept) | 0.989527 | 0.649926 | 1.522523 | 65 | 0.132729 | -0.30846 | 2.287518 |
| size | 0.002742 | 0.000118 | 23.30437 | 65 | < .0001 | 0.002507 | 0.002977 |
| MODEL_1 | -2.79206 | 0.735072 | -3.79835 | 65 | < .0001 | -4.2601 | -1.32402 |

Table 6. The fixed effects coefficients of the model explaining the tour cost in terms of the HPN+2opt/GPN+2opt and the problem size.

| Name | Estimate | SE | tStat | DF | pValue | Lower | Upper |
|---|---|---|---|---|---|---|---|
| (Intercept) | 17.96973 | 2.224075 | 8.079641 | 65 | < .0001 | 13.52795 | 22.41152 |
| size | 0.002559 | 0.000488 | 5.243273 | 65 | < .0001 | 0.001584 | 0.003534 |
| MODEL_1 | 0.724771 | 0.097485 | 7.434709 | 65 | < .0001 | 0.53008 | 0.919461 |

notion by adding the Euclidean distance between the currently selected city and other cities to the vector context.

The Euclidean distance between two points $(x_{i1}, x_{i2})$ and $(x_{j1}, x_{j2})$ is shown in (0.11):

$$d = \sqrt{\left(x_{i1} - x_{j1}\right)^2 + \left(x_{i2} - x_{j2}\right)^2} \tag{0.11}$$

Where $x_{i1}$, $x_{i2}$, $x_{j1}$ and $x_{j2}$ are the coordinates of city $i$ and city $j$ respectively.

Using the feature extractor, we train our large model in TSP50, validate with TSP500, 10 epochs, 1e-3 learning rate with leaning rate decay 0.96 and 100 for tanh clipping. Due to memory constraints we only solve 1k instances, some sample tours are shown in Fig 6, in which we solve TSP50-250-500-1000 with HPN+2opt. Table 2 summarizes our result, which shows that our model generalizes better than GPN. For the sake of a fair comparison with the state-of-the-art (i.e., GPN), we used 2opt local search technique to fine-tune the HPN's tours. As shown in Table 2 and Fig 7, our models outperform the GPN, GPN+2opt, PN, AM, and 2opt models. Moreover, HPN+2opt returns near-optimal tours and generalizes better than the GPN on a large-scale instance.

## 3. Benchmark instances results and statistical analysis

To validate our model against the standard benchmark instances, we employed varied-size instances from the public libraries TSPLIB [27] and World TSP. The benchmark dataset consists of 34 instances. The naming convention of instances consists of the first few letters of the instance location and the problem size n. For example, the instance eg7146 has 7146 points in Egypt. The instance sizes vary from 400 to 10639 nodes(cites). The normalized and actual tour length in km and the testing time in seconds are reported in Table 3.

To understand how HPN is performing compared to the GPN (the state-of-the-art network) in terms of the tour cost and testing time shown in Table 3, we did statistical comparison between these two networks. The statistical model should consider the dependency between the observations shown in Table 3.

In other words, we should realize that the tour cost of HPN and GPN for the same instance are correlated. Moreover, the testing times of same instance using the two networks are correlated as well. Therefore, we used LME regression to explain the variability in the tour cost and testing time (i.e. the responses) [24]. The LME regression model explains the variability of the tour cost and testing time as a function of the network used (i.e., HPN and GPN) and the size

Table 7. The fixed effects coefficients of the model explaining the testing time in terms of the HPN+2opt/GPN+2opt and the problem size.

| Name | Estimate | SE | tStat | DF | pValue | Lower | Upper |
|---|---|---|---|---|---|---|---|
| (Intercept) | -143.885 | 25.28218 | -5.69116 | 65 | 3.26E-07 | -194.377 | -93.393 |
| size | 0.106226 | 0.005288 | 20.08878 | 65 | < .0001 | 0.095665 | 0.116786 |
| MODEL_1 | 42.49765 | 15.33099 | 2.77201 | 65 | < .0001 | 11.87955 | 73.11574 |

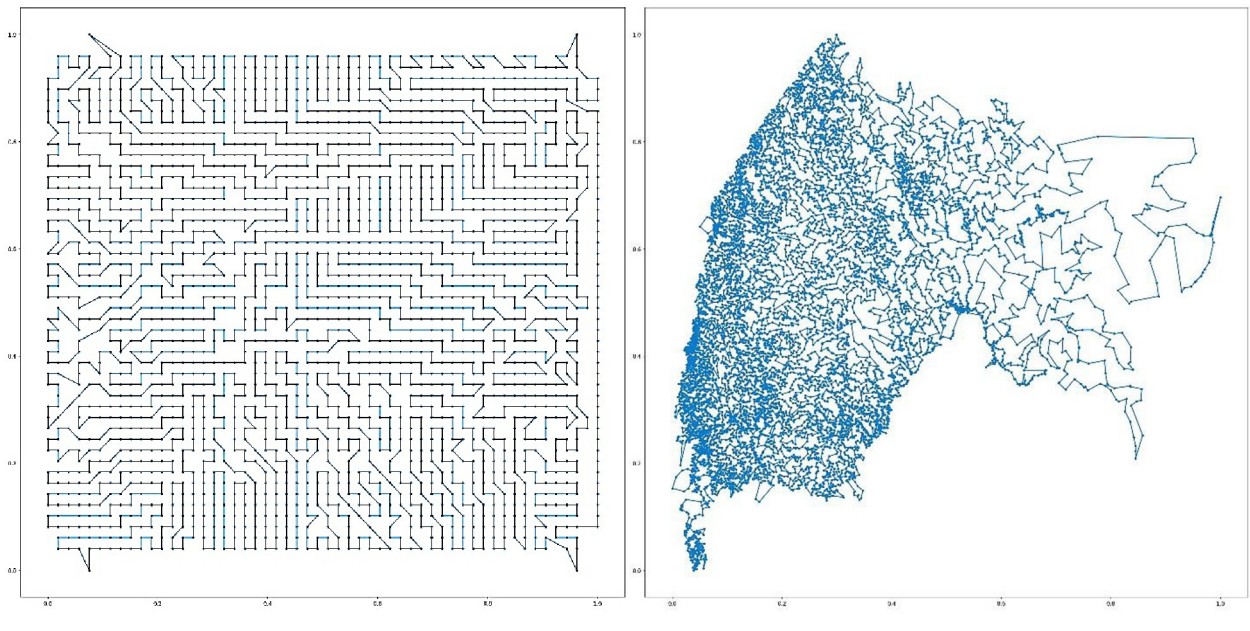

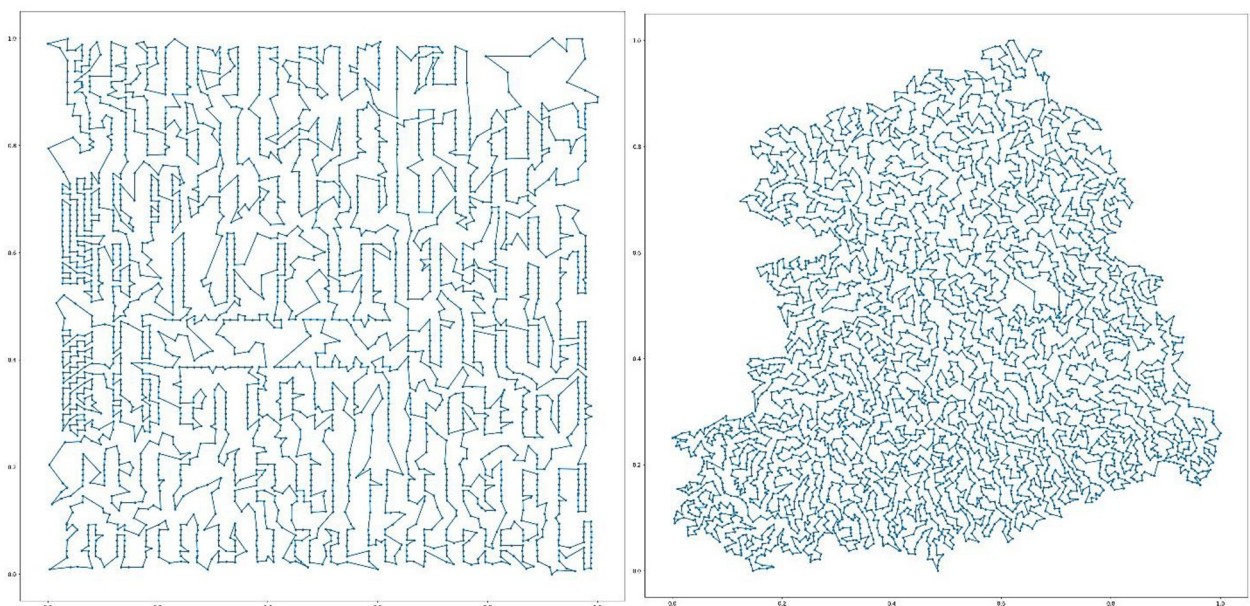

**Fig 8. Sample tours for benchmark instances.**

of the network. We used one indicator variable called "MODEL" to code the GPN and HPN.

$$MODEL = \begin{cases} 1 \ \textit{when the instance was solved using GPN} \\ 0 \ \textit{when the instance was solved using HPN} \end{cases}$$

In Table 4, we compared the tour cost for the HPN and GPN. A significance level of 0.05 was applied on all regression models. The p-value of the MODEL_1 indicator variable (i.e., GPN) is $< .0001$ and we conclude that the tour cost of the GPN is statistically significantly

higher than the tour cost of the HPN. However, as shown in Table 5, the testing time of the HPN is statistically significantly higher than the testing time of the GPN.

We repeat the same analysis for the tour cost and the testing time of the HPN+2opt and GPN+2opt. As shown in Tables 6 and 7, the HPN+2opt has statistically significantly lower testing time and tour cost. This is expected because HPN returns a better tour as an initial point in the solution space which helps 2opt to find a better final tour in less time. Finally, for the sake of completeness, we visualized four constructed tours using HPN + 2opt in Fig 8.

## VI. Conclusion and future work

In this work, a hybrid pointer network (HPN) is proposed for tackling both small-scale and large-scale problems. We demonstrate that the hybrid concept with a graph-based method is successful in improving the model performance for both scales. We used REINFORCE with Rollout Baseline to train our model. Our results show that our model outperforms the traditional graph pointer network with a significant margin, resulting in improved model generalization. Our model is still struggling to find the optimal solution for larger instances. Finding the optimal solution for larger instances is challenging so this will be our future direction.

Our future work, we will attempt to find a robust architecture to improve the quality and the time of solutions for large-scale problems, resulting in better model generalization. We also want to tackle combinatorial problems with constraints, which will be an important direction for future study.

## Acknowledgments

This research was supported (partially or fully) by the Motor Accident Insurance Commission (MAIC) Queensland.The views expressed herein are those of the authors and are not necessarily those of the MAIC.

## Author Contributions

**Conceptualization:** Ahmed Stohy, Heba-Tullah Abdelhakam, Sayed Ali, Mohammed Elhenawy, Abdallah A. Hassan, Mahmoud Masoud, Sebastien Glaser, Andry Rakotonirainy.

**Data curation:** Ahmed Stohy, Heba-Tullah Abdelhakam, Sayed Ali, Mohammed Elhenawy, Abdallah A. Hassan.

**Formal analysis:** Heba-Tullah Abdelhakam.

**Investigation:** Ahmed Stohy, Heba-Tullah Abdelhakam, Mohammed Elhenawy, Mahmoud Masoud, Andry Rakotonirainy.

**Methodology:** Ahmed Stohy, Sayed Ali, Mohammed Elhenawy, Abdallah A. Hassan, Mahmoud Masoud, Sebastien Glaser, Andry Rakotonirainy.

**Software:** Sayed Ali, Mohammed Elhenawy.

**Supervision:** Mohammed Elhenawy, Abdallah A. Hassan, Mahmoud Masoud, Sebastien Glaser, Andry Rakotonirainy.

**Validation:** Ahmed Stohy, Mohammed Elhenawy, Abdallah A. Hassan, Sebastien Glaser.

**Visualization:** Ahmed Stohy, Sayed Ali.

**Writing – original draft:** Ahmed Stohy, Heba-Tullah Abdelhakam, Sayed Ali.

**Writing – review & editing:** Mohammed Elhenawy, Abdallah A. Hassan, Mahmoud Masoud, Sebastien Glaser, Andry Rakotonirainy.

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
