## [Decision Letter · Decision Letter 0]

8 Nov 2021

PONE-D-21-33733Hybrid Pointer Networks for Traveling Salesman Problems OptimizationPLOS ONE

Dear Dr. Masoud,

Thank you for submitting your manuscript to PLOS ONE. After careful consideration, we feel that it has merit but does not fully meet PLOS ONE’s publication criteria as it currently stands. Therefore, we invite you to submit a revised version of the manuscript that addresses the points raised during the review process.

We look forward to receiving your revised manuscript.

Kind regards,

Seyedali Mirjalili

Academic Editor

PLOS ONE

Journal Requirements:

3. Your abstract cannot contain citations. Please only include citations in the body text of the manuscript, and ensure that they remain in ascending numerical order on first mention.

Reviewers' comments:

Reviewer's Responses to Questions

**Comments to the Author**

1. Is the manuscript technically sound, and do the data support the conclusions?

Reviewer #1: Yes

Reviewer #2: Yes

2. Has the statistical analysis been performed appropriately and rigorously? 

Reviewer #1: Yes

Reviewer #2: No

3. Have the authors made all data underlying the findings in their manuscript fully available?

Reviewer #1: No

Reviewer #2: Yes

4. Is the manuscript presented in an intelligible fashion and written in standard English?

Reviewer #1: No

Reviewer #2: Yes

5. Review Comments to the Author

Reviewer #1: The manuscript, in its present form, contains several weaknesses. Adequate revisions to the following points should be undertaken in order to justify the recommendation for publication.

1. Please re-write the abstract section. It is recommended that the reference not be in the abstract.

2. Please write a contribution of this method in the end of Introduction section.

3. Related work is very weak. There are many research ın the literature for solving TSP problem by meta-heuristic algorithms. This algorithm have been used a lot and have shown that they can get good performance. Therefore, it is suggested to add 2021 articles in the field of meta-heuristic in related work section.

4. It is recommended to explain the proposed model in a separate section. To be comfortable for readers. (Flowchart is also recommended for the convenience of readers)

5. What is the setting of the parameters? ٍ Explain this situations.

6. The authors should clearly state the limitations of the proposed method in real applications.

7. All the images are blurry, especially the graphs. Provide clearer images.

8. This paper has spelling and grammatical errors. Please fix all of them.

9. Display BKS values in tables.

10. As for the results, they should be commented and should lead to meaningful conclusions. At the moment, the manuscript is simply reporting some numerical values but consideration on the algorithmic behavior based on the obtained results are not adequately reported.

Reviewer #2: This manuscript is suitable for publication in this journal with the following revises, my recommendation is as follows:

1: the abstract section I too long, please delete unnecessary sentences.

2: What is your main contribution? Please highlight it in this manuscript.

3: How are the optimal parameters selected? It is well known that modeling complex dynamic systems are challenging. See the recent papers: Discrete farmland fertility optimization algorithm with metropolis acceptance criterion for traveling salesman problems. It can explain the study's method or indicate the contribution in the manuscript.

4: There is a Error in page 11 (Error! Reference source not found), please solve it.

5: You have used 2opt local search. What is the main reason for choosing this version of local search? If 3opt and LKH are also available.

6: What is your main purpose in using local search? Generate an iterated LOCAL SEARCH OR multi-start local search model or create a model that simulates both of these mechanisms.

7: It is recommended to re-examine and design the Figures.

8: Refer and explain the models used for comparison. For example, LKH-3 is software produced to solve the problems of the TSP and VRP.

9: Is Concorde the same as Concorde software? In this case, this software uses algorithms with different settings. Please explain these settings and the algorithm used.

10: Time criterion is the total execution time or the execution time to get the best solution? Clarify this case, it is vague.

11: Please refer to TSPLIB.

12: In evaluations, please use large instances for evaluation.

13: In the results, for example in instance (rd400), the value of Obj. shows 17. But in this example the value of Best known solution (BKS) is equal to 15,281. What exactly do you mean by these values?! The results are based on how many independent performances?

14: Please use PDav(%)=( Ave.-BKS)/BKS×100 for better evaluation and comparison.

6. PLOS authors have the option to publish the peer review history of their article (what does this mean?). If published, this will include your full peer review and any attached files.

Reviewer #1: No

Reviewer #2: No

---

## [Author Response · Author response to Decision Letter 0]

11 Nov 2021

Reviewer #1

1. Please re-write the abstract section. It is recommended that the reference not be in the abstract.

• Thank you! We handled this issue by removing any citations from the abstract.

2. Please write a contribution of this method at the end of the Introduction section.

• Thank you! We added the summary of our contribution at the end of the introduction.

3. Related work is very weak. There are many research in the literature for solving TSP problem by meta-heuristic algorithms. This algorithm has been used a lot and has shown that they can get good performance. Therefore, it is suggested to add 2021 articles in the field of meta-heuristic in the related work section.

• Thank you! We added a more recent method for TSP, and our main concern is in the RL-based models, so we didn't give much attention to meta-heuristic and any method built upon the integer programming, we target the RL-based models so we focused all our effort on that direction.

4. It is recommended to explain the proposed model in a separate section. To be comfortable for readers. (Flowchart is also recommended for the convenience of readers)

• Thank you! That's what we did, we made a separate section (section IV) to illustrate our model and figure 1 illustrate our model's operations graphically.

5. What is the setting of the parameters? ٍ Explain this situation.

• Thank you! It's common hyperparameters used in the literature of RL, and for the sake of fairness, we used the same settings that are used in the paired methods to illustrate that our architecture is the one that improves the performance.

6. The authors should clearly state the limitations of the proposed method in real applications

• Thank you! We added the model's limitation at the end of the conclusion section

7. All the images are blurry, especially the graphs. Provide clearer images.

• Thank you! Handled.

8. This paper has spelling and grammatical errors. Please fix all of them.

• Thank you! Handled

9. Display BKS values in tables.

• Thank you! Our main concern in this study to prove that our model improves the performance of the GPN so we only record the performance for two algorithms to Attract the reader's attention into that point

10. As for the results, they should be commented and should lead to meaningful conclusions. At the moment, the manuscript is simply reporting some numerical values but consideration on the algorithmic behaviour based on the obtained results are not adequately reported.

• Thank you! Handled

Reviewer #2

1. The abstract section I too long. Please delete unnecessary sentences.

• Thank you! Handled 

2. What is your main contribution? Please highlight it in this manuscript.

• Thank you! We added the summary of our contribution at the end of the introduction.

3. How are the optimal parameters selected? It is well known that modelling complex dynamic systems are challenging. See the recent papers: Discrete farmland fertility optimization algorithm with metropolis acceptance criterion for travelling salesman problems. It can explain the study's method or indicate the contribution in the manuscript.

• Thank you! It's common hyperparameters used in the literature of RL and for the sake of fairness, we used the same settings that are used in the paired methods to illustrate that our architecture is the one that improves the performance.

4. There is an Error in page 11 (Error! Reference source not found), please solve it.

• Thank you! Handled

5. You have used 2opt local search. What is the main reason for choosing this version of local search? If 3opt and LKH are also available.

• Thank you! For the sake of fairness because we compared our model with GPN, and the author used the 2-opt to improve the quality of the solution

6. What is your main purpose in using local search? Generate an iterated LOCAL SEARCH OR multi-start local search model or create a model that simulates both of these mechanisms.

• Thank you! For the sake of fairness because we compared our model with GPN, and the author used the 2-opt to improve the quality of the solution

7. It is recommended to re-examine and design the Figures.

• Thank you! Handled

8. Refer and explain the models used for comparison. For example, LKH-3 is software produced to solve the problems of the TSP and VRP.

• -Thank you! Handled

9. Is Concorde the same as Concorde software? In this case, this software uses algorithms with different settings. Please explain these settings and the algorithm used.

• Thank you! We took the result as it is from the GPN's paper, so we didn't know the setting used for solving the instances

10. Time criterion is the total execution time or the execution time to get the best solution? Clarify this case, it is vague.

• - Thank you! The time that the model took for solving the 10k instances 

11. Please refer to TSPLIB.

• Thank you! Handled.

12. In evaluations, please use large instances for evaluation.

• Thank you! Handled.

13. In the results, for example, in the instance (rd400), the value of Obj. shows 17. But in this example, the value of the Best-known solution (BKS) is equal to 15,281. What exactly do you mean by these values?! The results are based on how many independent performances?

• Thank you! In the first two columns, we report the normalized distance, we downscale the coordinate between 0 and 1 by this way, the model trains fast 

14. Please use PDav(%)=( Ave.-BKS)/BKS×100 for better evaluation and comparison.

• Thank you! Our main concern in this study to prove that our model improves the performance of the GPN so we only record the performance for two algorithms to Attract the reader's attention into that point

---

## [Decision Letter · Decision Letter 1]

22 Nov 2021

Hybrid Pointer Networks for Traveling Salesman Problems Optimization

PONE-D-21-33733R1

Dear Dr. Masoud,

We’re pleased to inform you that your manuscript has been judged scientifically suitable for publication and will be formally accepted for publication once it meets all outstanding technical requirements.

Kind regards,

Seyedali Mirjalili

Academic Editor

PLOS ONE

Additional Editor Comments (optional):

Reviewers' comments:

Reviewer's Responses to Questions

**Comments to the Author**

1. If the authors have adequately addressed your comments raised in a previous round of review and you feel that this manuscript is now acceptable for publication, you may indicate that here to bypass the “Comments to the Author” section, enter your conflict of interest statement in the “Confidential to Editor” section, and submit your "Accept" recommendation.

Reviewer #1: All comments have been addressed

Reviewer #2: All comments have been addressed

2. Is the manuscript technically sound, and do the data support the conclusions?

Reviewer #1: Yes

Reviewer #2: Yes

3. Has the statistical analysis been performed appropriately and rigorously? 

Reviewer #1: Yes

Reviewer #2: Yes

4. Have the authors made all data underlying the findings in their manuscript fully available?

Reviewer #1: Yes

Reviewer #2: Yes

5. Is the manuscript presented in an intelligible fashion and written in standard English?

Reviewer #1: Yes

Reviewer #2: Yes

6. Review Comments to the Author

Reviewer #1: Dear Editor, More thanks for you, I read this paper, authors done all my comments, so in my opinion , this paper accept for publication.

Reviewer #2: In my perspective Paper(Hybrid Pointer Networks for Traveling

3 Salesman Problems Optimization) is accepted from my side.

7. PLOS authors have the option to publish the peer review history of their article (what does this mean?). If published, this will include your full peer review and any attached files.

Reviewer #1: No

Reviewer #2: No

---

## [Editor Report · Acceptance letter]

3 Dec 2021

PONE-D-21-33733R1 

Hybrid Pointer Networks for Traveling Salesman Problems Optimization 

Dear Dr. Masoud:

I'm pleased to inform you that your manuscript has been deemed suitable for publication in PLOS ONE. Congratulations! Your manuscript is now with our production department. 

Kind regards, 

on behalf of

Prof. Seyedali Mirjalili 

Academic Editor

PLOS ONE